# Comprehensive Review of Recent Research Advances on Flame-Retardant Coatings for Building Materials: Chemical Ingredients, Micromorphology, and Processing Techniques

**DOI:** 10.3390/molecules28041842

**Published:** 2023-02-15

**Authors:** Fang-Fang Li

**Affiliations:** China Railway First Survey and Design Institute Group Co., Ltd., Xi’an 710043, China; lifangfang.dr@gmail.com

**Keywords:** building materials, fire-retardant coatings, layer-by-layer self-assembly, hot pressing, nanosheets, graphene oxide

## Abstract

Developing fire-retardant building materials is vital in reducing fire loss. The design and preparation of novel fire-retardant coatings merely require the adhesion of flame retardants with high fire-retardant characteristics on the surface, which is significantly more economical than adding excessive amounts of flame retardants into bulk building materials. Meanwhile, fire-retardant coating has excellent performance because it can block the self-sustaining mechanisms of heat and mass transfer over combustion interfaces. In recent years, research of fire-retardant coatings for building materials has been subject to rapid development, and a variety of novel environmentally benign fire-retardant coatings have been reported. Nonetheless, as the surface characteristics of various flammable building materials are contrastively different, selecting chemical ingredients and controlling the physical morphology of fire-retardant coatings for specific building materials is rather complicated. Thus, it is urgent to review the ideas and preparation methods for new fire-retardant coatings. This paper summarizes the latest research progress of fire-retardant building materials, focusing on the compositions and performances of fire-retardant coatings, as well as the principles of their bottom-up design and preparation methods on the surface of building materials.

## 1. Introduction

Frequent indoor fires cause massive damage and injuries, leading to enormous loss of life worldwide. In 2020 alone, among 3.3 billion people in 48 countries surveyed, 20.6 thousand people died in fire disasters, and 69.5 thousand people were injured [1]. Furthermore, in recent decades, urbanization in nations with major populations has led to massive densification of houses [2]. Meanwhile, flammable but high-performance polymers and woods are increasingly adopted as façades and decorations for buildings [3]. In this context, fire risks are significantly aggravated. As such, it is urgent to design and optimize the structure and composition of building materials to reduce their flammability.

In principle, the combustion of flammable building materials is based on a self-sustainable mechanism of heat and mass transfers at the interface between the burning substrate and flames [4]. Organic polymers in flammable materials decompose to generate volatile small molecules and free radicals at high temperatures, which are mixed with air in the gaseous phase. Then, the emitting organics in the air are further combusted to release heat [5]. Thereby, the heat feedback mechanism prompts combustion until oxygen or the material is exhausted. Flame-retardant materials virtually block the self-sustainable mechanism of combustion, including two aspects: (i) stopping heat transfer from the gaseous phase into flammable materials and (ii) insulating mass transfer from flammable materials into the gaseous phase [6]. For example, halogen-containing flame retardants are highly efficient and conventionally most used in building materials because they meet both requirements. Specifically, such flame retardants quickly decompose and absorb heat from combustion, emitting nonflammable molecules, such as halogen hydrides, to quench radical chain reactions [7]. However, concerning the toxicity and waste management regulations in many countries, halogen-containing flame retardants are currently unfavorable and even banned from use in building materials [8,9,10]. Instead, halogen-free flame retardants containing phosphorous or boron have been explored as alternatives [11,12]. Unlike halogen-containing flame retardants, which block combustion progress in the gaseous phase, the new flame retardants focus on stopping combustion in the solid phase. Specifically, flame retardants are thermally stable, and their decomposition absorbs large amounts of heat. Nonetheless, to insulate heat and mass transfer across the combustion interface, it is required that such flame-retardant materials be quickly transformed into a coke bed with intumescent structures or compact surfaces in the solid phase. Therefore, it is mandatory to add high contents of such halogen-free flame retardants into bulky materials to reduce their flammability (e.g., 25 w% additive of phosphorous flame retardant) [13]. The high loading of flame retardants as additives increases costs and raises compatibility issues. In addition, doped flame retardants often deteriorate the appearance and mechanical strength of building materials [14,15,16]. Moreover, increased usage of flame-retardant additives and their continuous release from building materials into indoor environments have been found to be persistent, accumulative, and hazardous to human health [17,18,19].

In lieu of adding flame-retardant additives into bulky building materials, the design and preparation of a thin coating of flame retardants should be more effective since the combustion progress is virtually retarded on the interface between the solid and gaseous phases [20]. Therefore, the usage and cost of flame retardants can be significantly reduced, while the coatings do not alter the mechanical strength and color of building materials. Recently, eco-benign, flame-retardant coatings for building materials have been subject to rapid development. A large variety of novel organic and inorganic composites, as well as their preparation techniques, has been reported. Nonetheless, surface properties of multifarious flammable building materials, such as wood, plastics, and textiles, are rather diverse. Thus, in the design of flame-retardant coating, it is crucial to attain appropriate compatibility between the coating and bulk material to facilitate tight adhesion between the two phases. Accordingly, in contrast to the selection and addition of flame-retardant additives, the chemical composition, physical micromorphology, and processing technique should obey the bottom-up design strategy. Here, the chemical composition, physical micromorphology, and processing techniques of novel flame-retardant coatings reported in recent years are summarized. The general design principles of flame-retardant coatings on various building materials are concluded to prompt their large-scale production and industrial application.

## 2. Surface Properties of Flammable Building Materials

Wood is a sustainable building material, exhibiting high endurance, low density, high strength, super toughness, and great malleability. As a traditional building material, ancient architecture is mostly built with wood. In terms of chemistry, woods mainly comprise cellulose and lignin. While cellulose bears abundant hydroxyl groups, providing woods with advanced toughness, lignin is a group of highly complex phenolic polymers composed of coumarin, coniferol, and sinapinol, endowing woods with high strength (Figure 1a). Under conditions of fire, woods carbonize and release flammable gases, such as carbon monoxide, methane, and tar, to sustain the combustion progress [21]. Carbonized fibers of wood are easily split along textures, allowing for rapid transfer of mass and heat across the combustion interface. As such, the combustion region spreads downward from the surface into the substrate very quickly. Considering the presence of highly abundant hydroxyl and phenyl moieties on the surface of woods, flame-retardant coatings could be attached through hydrogen bonding and π-π stacking.

Plastics, which are polymers derived from the downstream products of the petroleum industry, are widely employed building materials. These materials are generally unstable under heat and quickly decompose under fire conditions, emitting substantial amounts of smoke. Typical building plastics, including polyurethane, polystyrene, acrylonitrile–butadiene–styrene (ABS) resins, etc., have hydrophobic surfaces (Figure 1b–d). Among building plastics, polyurethane and polystyrene are widely employed to manufacture insulation foams. Due to their loose and sometimes fluffy structures, foams are highly flammable. In addition, the decomposition products of flammable plastics are even more hazardous. For example, polyurethane decomposes and releases toxic and suffocating gases of isocyanates and carbon dioxide beyond 300 °C. Meanwhile, catalytical radicals of imines and alkenes are produced, further enhancing its decomposition [22]. ABS resins are copolymers of acrylonitrile, butadiene, and styrene, which are highly hydrophobic. ABS resins are thermoplastic and widely employed as building materials for air outlets, wires, and cables. Similar to polyurethane, ABS resins release toxic and suffocating gases, including carbon monoxide, carbon dioxide, hydrogen cyanide, etc. [23].

Textile materials, including carbon fibers, cellulose, nylon, aramid fiber cloth, etc., are typically used for modern building construction [24,25,26,27]. Except for aramid fibers, most textile building materials are fabricated with aliphatic polymers, giving the materials low strength, high malleability, and excellent toughness (Figure 1e). On the other hand, aramid fiber cloth contains abundant phenyl moieties, showing abrasion resistance. Therefore, aramid fiber cloth is often adopted as reinforcement in composites, such as concretes (Figure 1f) [28]. However, high-performance textile building materials are also highly flammable due to their polymeric carbon skeleton and large surface area.

In conclusion, flammable building materials typically contain thermally unstable polymers that easily undergo pyrolysis through radical chain reactions under fire conditions. Moreover, these products cannot form compact coke. Thus, they are unable to block mass and heat transfer from the surface into the bulky body of flammable building materials. On the other hand, both the performance and appearance of building materials need to be considered in architectural design. Therefore, it is unfavorable for the appearance, endurance, or mechanical properties of building materials to be altered by adding flame-retardant materials. To meet these needs, minimal amounts of flame-retardant materials should be coated on the surface of substrates to effectively stop a fire. Most flammable building materials bear hydroxyl, amino, and imine moieties on the surface. Meanwhile, low surface coverages of such polar functional groups result in a hydrophobic surface of building materials. Considering the surface properties of all the discussed building materials, it is envisaged that the chemistry of hydrogen bonding, coordinate bonding, and hydrophobic interactions is essential to afford a strong affinity between the surface of building materials and flame-retardant coating materials (e.g., epoxy, polyurea, polydopamine, etc.).

## 3. Chemical Composition and Physical Micromorphology of Flame-Retardant Coatings

### 3.1. Inorganic Materials in Flame-Retardant Coatings

Natural minerals used as flame-retardant coatings include montmorillonite, kaolinite, vermiculite, brucite, magnesite, wollastonite, bentonite, etc. [29,30,31,32,33,34]. Alkalic minerals can absorb harmful acidic gases released by flammable building materials under fire. Meanwhile, minerals are highly thermally stable. Their decomposition under extremely high temperatures not only absorbs a significant amount of heat but also generates nonflammable gases to dilute radicals and flammable gases, blocking the mass and heat feedback mechanism of combustion. The most favorable flame-retardant minerals are monoclinic crystalline, containing more than two metal ions. They typically have an intumescent layered micromorphology [30,35,36,37]. For example, montmorillonite contains layered structures of aluminum and magnesium silicates [30]. Bentonite has a sandwiched microstructure comprising two layers of tetrahedron silicon oxide and one layer of octahedron aluminum oxide [36]. Vermiculite also exhibits layered microstructures containing magnesium, aluminum, iron, and silicon, which is intumescent and peels off the surface of building materials under fire [37]. In addition, natural minerals such as attapulgite and halloysite are nanotubes or nanorods, which can easily disperse in flame-retardant coatings [38,39].

Inorganic minerals are relatively difficult to attach to the surface of building materials since the surfaces of materials often bear fewer ionizable moieties to form ionic bonds. Meanwhile, inorganic minerals are unable to form hydrogen bonding or hydrophobic interactions on the surface. Therefore, organic media such as graphene or polymers are added to bridge inorganic minerals and surface moieties of flammable building materials. In such a design, the dispersity of minerals in the organic medium is a crucial factor determining the efficiency of flame-retardant coatings. Unfortunately, except for the aforementioned natural minerals showing well-ordered micromorphology, most natural minerals are bulky crystalline and unable to be suspended in organic media. As such, the minerals need to be mechanically crushed, ground, and appropriately sieved to prepare flame-retardant coatings [40,41]. For example, Ahmad introduced up to 4 w% ground wollastonite into intumescent flame-retardant coating materials, increasing char expansion by 34%. Meanwhile, it was found that a longer grinding time and a higher amount of the ground wollastonite ingredient improved the thermal properties of the intumescent coating [40].

Recently, minerals from waste or recyclable resources have been explored in preparing flame-retardant coatings [42]. For example, Yiew et al. demonstrated the preparation of an intumescent flame-retardant coating using vinyl acetate copolymer doped with 3 w% nanofillers of eggshell [43]. Additionally, bio-ashes containing abundant silicates, oxides, and hydroxides could also be used as fillers to prepare intumescent flame-retardant coatings [44,45]. Bio-ashes are favorable for use as alternatives to natural minerals for the preparation of flame-retardant coatings since they can be easily ground into fine powders and dispersed well in organic media. Abdullah et al. found that rice husk ash as an ingredient increased the total and open porosities and rough surfaces of the coating material, improving its intumescent flame-retardant performance [44]. Moreover, Rybinski et al. took advantage of the high surface area and cenospheric morphology of fly ash to adsorb a nanoiron catalyst onto its surface. The nanocomposite was prepared as nanofillers in flame-retardant coatings, increasing the LOI value of butadiene–acrylonitrile rubber substrates from 23% to 28.2%. In addition, the loaded iron could also catalyze the carbonation of polymers during combustion to form a coke bed on the surface and rapidly stop fires [46].

### 3.2. Organic Molecules in Flame-Retardant Coatings

Phytic acid, polyphenol, and polydopamine are the most employed biological molecules in flame-retardant coatings [47,48]. Phytic acid, also known as inositol hexaphosphate, contains P-O-C units that are highly stable and not thermally decomposed in combustion. Meanwhile, phytic acid has multiple phosphorous acid groups to chelate transitional metal ions, forming layered structures. Thus, flame-retardant coatings containing phytic acid easily form compact coke layers on the surface of materials in combustion, blocking mass and heat transfer and the self-sustaining mechanism of fires. Nonetheless, the acidity of phytic acid is too strong to be compatible with the relatively hydrophobic surfaces of most building materials. As such, alkalic molecules or polymers are often added to bridge phytic acids on the surface of materials. Pan et al. assembled a three-layer sandwiched structure of flame-retardant coating on polyvinyl alcohol by inserting one layer of phytic acid between two layers of melamine-doped polyethylene imine. The coating material is ultrathin and transparent, accounting for merely ~6 w% of the polyvinyl alcohol (PVA) substrate. Moreover, it reinforced the tensile strength of PVA and significantly lowered its peak heat release rate (pHRR) by 37% under fire [49]. Li et al. constructed a flame-retardant coating of 30 layers of alternatively arranged anionic phytic acid and cationic layers of poly [3-(5,5-cyanuricacidpropyl)-siloxane-co-trimethyl ammonium propyl siloxane chloride] [50]. Its flame-retardant performance on the surface of cotton fabrics was evidenced by an elevated limiting oxygen index (LOI) by 29.8%. The flaw of phytic acid as a coating material is that the molecule is virtually acidic and washable by water. Guo et al. designed a flame-retardant layer with cationic alkylammonium functional silsesquioxane (A-POSS) and anionic phytic acid (PA) complexed through ionic forces and adsorbed through hydrogen bonding onto the surface of cotton fabrics. The ionic coating cannot sustain rain wash. To cope with this problem, a superhydrophobic layer composed of titanium-oxide-doped polydimethylsiloxane (PDMS) was assembled on top. The hydrophobic PDMS adhered tightly to the surface of cotton fabrics, while titanium was chelated with phytic acid to form a compact surface of the coating materials. The coating efficiently increased the LOI of pristine cotton fabrics from 18% to 29% [51].

Tannic acid is one of the most favorable natural polyphenols in flame-retardant materials because the molecule easily carbonizes in combustion. Meanwhile, its gallic acid units can scavenge radicals produced by polymer decomposition under fire [52,53]. Moreover, tannic acid decomposed by heat releases nonflammable gases, such as phenylene triol and carbon dioxide. Therefore, flame-retardant coatings containing tannic acid are typically intumescent. During combustion progress, the intumescent coke layer formed on the surface of materials can efficiently block mass and heat transfer across the burning interfaces [54]. On the other hand, the molecular network of tannic acid contains abundant phenyl units and hydroxyl groups. Thus, it can easily adhere to the surface of most building materials through hydrophobic interactions and hydrogen bonding. Deniz et al. synthesized flame-retardant coatings through the reaction between tannic acid and hexachlorocyclophosphazene (HCCP) (Figure 2) [52]. The two molecules interconnect to form an organic layer enriched with P-O-C bonds, which are stable in combustion, promoting the formation of insulating coke beds on the surface of flammable materials. The coating material, accounting for 12 w% of the substrate, significantly elevated the LOI value of cotton to 35%. Recently, Ramirez compared the flame-retardant performance of tannic acid of different molecular weights (i.e., medium molecular weight, 2986 Da and 5573 Da). The flame-retardant coating containing higher-molecular-weight tannic acid was more intumescent. Nonetheless, the coating materials composed of low-molecular-weight tannic acid showed a higher carbonization index, less mass loss, and better flame-retardant efficiency [55].

Polydopamine contains abundant catechol moieties and is well-known for its strong adhesion towards surfaces of both organic and inorganic materials. As such, it is widely employed for the surface modification of building materials. For example, Li et al. coated polydopamine onto the surface of polystyrene that had hydrophobic carbon skeletons without specific functional groups. In combustion, the polydopamine coating formed compact coke layers, reducing the pHRR by 18.4% and restricting the release of flammable volatiles [56]. Meanwhile, polydopamine can scavenge radicals and quench chain reactions that induce polymer decomposition. Simultaneously, polydopamine is a versatile adhesive agent that can immobilize other less adhesive flame-retardant reagents in the organic media of coating materials. For example, minerals and polyacids are efficient flame-retardant materials. However, they are polar and generally incompatible with the hydrophobic surface of most building materials. In such cases, polydopamine can chelate metal ions in inorganic minerals and complex with polyacids through hydrogen bonding.

Liu et al. fabricated flame-retardant coating on the superhydrophobic surface of a polyurethane sponge by the adhesion of layered double hydroxide (i.e., Zn(OH)_2_ and Al(OH)_3_) with polydopamine. The coating material showed outstanding durability under harsh physical and chemical conditions. Meanwhile, its flame-retardant performance was evidenced by the increased LOI value of the neat polyurethane sponge from 18.5% to 24% [57]. Wang et al. also synthesized a strong flame-retardant coating by complexing polydopamine and polyphosphate through multiple hydrogen bonding. The coating layer, accounting for 13.7 w% of the substrate, efficiently increased the LOI value of pure cotton from 17.5% to 26.5% [58].

The molecular structures of tannic acid and polydopamine feature abundant polyphenol moieties. It is thus envisaged that flame-retardant coatings can be prepared by other natural polyphenols, such as ferulic acid, tea polyphenol, caffeic acid, chlorogenic acid, gallic acid, etc. [59]. However, polydopamine contrasts with other natural polyphenols because of its alkalinity. The polydopamine layer provides an alkaline local environment, allowing for the deposition of nonflammable metal oxide nanocrystalline material. The fine nanomorphology and large specific surface area of such nanocrystalline material can adsorb air, blocking mass and heat transfer across the interface between solid and gaseous phases (Figure 3) [60].

### 3.3. Layered Nanomaterials in Flame-Retardant Coatings

Materials showing layered micromorphology are widely employed in the design of novel flame-retardant coatings, including graphene nanosheets, hexagonal boron nitride platelets, montmorillonite nanosheets, MXene nanosheets, etc. [61,62,63,64,65]. Due to their excellent dispersity, nanosheets can be dispersed evenly in the organic media of coating materials. Meanwhile, nanosheets can serve as templates to deposit other flame-retardant ingredients, producing layer-by-layer assembled nanostructures. Such mimic-nacre nanostructures can be easily transformed into multiple-layer coke beds with compact surfaces to stop heat and mass transfer in combustion. In addition, two-dimensional layered nanomaterials typically show anisotropic heat conduction efficiency. Thus, the heat produced by combustion can be rapidly conducted along the surface of burning materials, reducing heat flow from the surface into the body [66]. Consequently, a meager amount of such nanosheets can significantly improve the flame-retardant efficiency of coating materials.

Graphene oxide nanosheets are widely employed nanomaterials in flame-retardant coatings. These materials contain a hydrophobic carbon skeleton enriched with hydroxyl and carboxylic functionalities. Thus, graphene oxides can adhere to surfaces of building materials through hydrophobic interactions and hydrogen bonding. Flame-retardant coatings made with graphene oxides alone show modest flame-retardant performances. Ji et al. prepared a flame-retardant coating consisting of nine layers of reduced graphene oxides on the surface of silk fabrics, which showed fire resistance and self-extinguished merely 5 s after ignition [67]. Moreover, graphene oxide contains carboxylic groups, which can complex with cationic polymers or minerals through ionic bonds at suitable pH, forming compact layer-by-layer assemblies. Carosio et al. assembled cationic poly(diallyldimethylammonium) in two layers of graphene oxide nanoplatelets. After grafting the three-layer coating on highly flammable polyurethane foams, the flexibility of foams was maintained, but the materials became self-extinguishing, nonignitable, and fire-resistant [68]. Maddalena et al. assembled cationic deacetylated chitosan between two graphene oxide layers. The three-layer coating grafted on polyurethane foams accounted for merely ~10 w% of the substrate but substantially improved the flame-retardant properties, reducing the pHRR by 54% and restricting smoke release by 59% [69]. The nanoplatelets of graphene oxide are well-dispersed in solutions of other ingredients. Therefore, this material can be employed to directly synthesize flame-retardant nanocomposites with uniform chemical compositions. Wang et al. mixed graphene oxide into a solution of carboxymethylcellulose containing a suspension of montmorillonite nanosheets (i.e., 16–22 μm and aspect ratio of 200–400). Instead of the layer-by-layer assembly (LBL) technique, which is complicated and time-consuming, the mixture facilely formed ternary mimic-nacre nanocomposites by the hot-pressing technique, exerting excellent self-extinguishing performances [63]. Graphene oxide also has a large specific surface area and abundant reactive groups to anchor other flame-retardant reagents. Li et al. chemically immobilized 9,10-dihydro-9-oxa-10-phosphaphenanthrene-10-oxide (DOPO) onto the surface of graphene oxide nanosheets through a Mannich reaction and silanization reaction. The material coated onto polystyrene increased its LOI from 18% to 29% [70]. Moreover, Xie et al. functionalized hydroxyl groups of cellulose with pyrimidine moieties, which can form strong ionic bonds with carboxylic moieties on graphene oxide. As such, tight layer-by-layer mimic-nacre coatings were facilely produced. The prepared coatings had compact layered nanostructures and significantly increased the LOI values of many flammable building materials, such as polypropylene, wood, and polyurethane foams, from 18%, 26.5%, and 17.5% to 31%, 41.5%, and 25.5%, respectively [71].

In combustion, graphene oxide nanoplatelets carbonize to produce compact graphene layers. Meanwhile, the conversion of graphene oxide to graphene results in a distinguishable increase in electrical conductivity. As such, graphene oxide can be used to design intelligent flame-retardant coatings with early fire warning functions [72,73,74,75]. Zhang et al. prepared multiple layers of graphene oxide using the LBL technique by filling molecules of L-ascorbic acid (LAA) and 3-methacryloxypropyltrimethoxysilane (MPMS) between layers (Figure 4) [72]. In combustion, LAA rapidly reduced graphene oxide into graphene, and MPMS crosslinked to form a compact graphene layer. SEM images show that the thin coating sheet exhibited a wrinkled structure, containing evenly distributed needle crystalline of LAA crystalline and tightly aligned graphene oxide sheets (Figure 4c–e). The transformation of such coating materials resulted in a prompt elevation of electrical conductivity within only 1 s to trigger fire alarms. It is worth mentioning that a loose assembly of graphene oxide layers is prepared in such a design. This is in contrast to conventional designs of flame-retardant coatings with the aim of synthesizing compact layered nanostructures. The loose assembly is favorable to fill large amounts of small reactive molecules. Moreover, the apparent difference in both chemical composition and micromorphology between the coating materials before and after combustion is pivotal in designing intelligent fire-alarm coatings.

Similar to graphene oxide nanoplatelets, hexagonal boron nitride (h-BN) nanoplatelets are also well-dispersed in aqueous solutions and are thermally stable. In addition, h-BN coatings are often colorless and transparent [76]. Therefore, they are more suitable than graphene oxide materials for combinations with building materials, such as woods and textiles. h-BN can be deposited on the surface of building materials through hydrogen bonding and coordination bonding (i.e., boron–oxygen–carbon units formed between the empty electron orbit of boron and the oxygen of dehydrogenated hydroxyl groups on the surface of building materials) [77]. Gan et al. made a coating of h-BN simply by painting its aqueous solution onto the surface of woods. After hot pressing, the coating materials with a 30 μm thickness were proven flame-retardant by a doubled ignition delay time and a 25% reduction in the maximum heat release rate of woods [37]. Davescne et al. made flame-retardant coatings with h-BN nanoplatelets and polyethylene imine; coordination bonds between the boron of h-BN and nitrogen of imine enabled the formation of compact double layers (Figure 5) [61]. Although the coating material accounted for merely ca. ~6.8 w% of the substrate (i.e., PU form), it effectively reduced its pHRR value by more than 50%. X-ray mapping, SEM, and TEM images further showed that the thin coating material maintained the open-cell structure of the PU foam substrate in the char after combustion (Figure 5c–e). Moreover, the coating material is transparent and can also absorb ultraviolet irradiance to protect the building materials against weathering and UV-induced aging (Figure 5f–g). Qiu et al. made flame-retardant coatings on flexible polyurethane foams with h-BN nanosheets and polyethylene imine through the LBL approach. The coating material accounting for 6.1~14.4 w% of the PU foam substrate effectively reduced the pHRR value by 50.1% and restricted CO release by 53.8% [78]. In addition, Liu et al. prepared flame-retardant coatings on cotton with h-BN nanosheets and hexachlorocyclotriphosphazene (HCCP), a typical inorganic flame-retardant reagent. Nonetheless, because there was no effective coordination between the two molecules, they were unable to form compact flame-retardant layers, and the prepared coating materials showed marginal flame-retardant efficiency by increasing the LOI of cotton fabrics from 20.1% to 24.1% [79]. It can be concluded that although h-BN has a well-defined nanomorphology, it is still mandatory to select compatible chemical reagents to form compact layers of composites to enhance its flame-resistant performance.

Recently, emerging two-dimensional (2-D) materials, such as MXenes, have been introduced to prepare flame-retardant coatings. MXenes are metal ion carbides or nitrides. Their formulas are represented as M_x_X_y_T_z_, where “M” indicates a transitional metal ion, “X” indicates carbon or nitrogen atoms, and “T” represents functional groups, e.g., -OH, -O, or -F [80]. Similar to graphene, MXene contains a large 2D surface area of conjugated double bonds. Thus, it exhibits good electric conductivity and mechanical properties. Compared to graphene, transitional metal ions on MXene, such as titanium, can catalyze the carbonization of polymers during combustion, inducing the rapid formation of coke beds to block heat and mass transfer and restrict the release of smoke [81]. Although exfoliated MXene has nanosheet morphology and is well-dispersed in aqueous solutions, it is highly flexible and self-agglomerable. Huang et al. employed 2-isocyanatoethyl methacrylate to functionalize hydroxyl groups of exfoliated MXene nanosheets to attenuate hydrogen-bonding-induced self-agglomeration and improve its dispersity in solutions. The chemically modified MXene nanosheets showed excellent film-forming performance on the surface of rigid polyurethane foams. By introducing merely 2 w% of such coating materials onto the substrate thermoplastic PU, the coating materials rapidly generated char and MXene jammed networks, blocking heat and mass transfer and restricting the release of volatile products in combustion [82]. Besides flame-retardant efficiency, MXene also possesses electromagnetic interference-shielding properties and high electrical conductivities and can therefore be applied in the preparation of multifunctional coatings. For example, a low content of 5.2 mg/cm^2^ MXene (i.e., Ti_3_C_2_T_x_) on the surface of cotton fabrics not only showed flame resistance by increasing the LOI from 19% to 36.5% but also exerted a high electromagnetic interference-shielding effectiveness of 31.04 dB [83].

### 3.4. Nanotubes and Nanorods in Flame-Retardant Coatings

Natural minerals such as halloysite and attapulgite are nanotubes or nanorods, which are well-dispersed in the media of coating materials to enhance their flame-resistant performances [84,85]. Recently, synthetic nanotubes have attracted increasing interest in the design of flame-retardant coatings. Nosaka et al. investigated the flame-resistant efficiency of amine-functionalized multiwall carbon nanotubes (NH_2_-MWCNT) on the surface of polyester fabric. Compared to amorphous carbon black, which lacks dispersity in solutions, NH_2_-MWCNT added to the coating materials showed distinguished flame-retardant efficiency and increased the remaining length of residues after burning. The amount of NH_2_-MWCNT loaded in the coating materials was highly correlated with their flame-resistant efficiency [86]. In addition, titanium oxide (TiO_2_) nanotubes show even higher thermal stability than carbon nanotubes, catalyze the carbonization of substrate materials during combustion, and absorb flammable volatiles. Nonetheless, the dispersity of TiO_2_ nanotubes in organic media is rather poor. To solve this problem, Pan et al. loaded TiO_2_ nanotubes on phosphatized chitin through coordination bonds between phosphorous moiety and titanium. Only 2 w% addition of such a strong composite in coating materials on epoxy resins showed remarkable flame-resistant efficiency by reducing the pHRR and total heat release by 48.8% and 42.1%. Meanwhile, the added TiO_2_ nanotubes promoted the formation of a compact char layer to retard heat and mass transfer [87]. Organic nanotubes prepared through Pd(0)/Cu(I)-catalyzed Sonogashira–Hagihara cross-coupling reaction of ethylnylbenes represent another class of organic nanotubes that show excellent flame-resistant efficiency. Wei et al. selected 1,3,5-triethynylbenzene and 1,4-diethynylbenzene as reagents for cross coupling into a hydrophobic layer, which rolled up into nanotubes with a width of 6.4 nm and a high specific surface area of up to 1194 m^2^g^−1^ (Figure 6 FCMP-1). In contrast, linear polymers synthesized from 1,4-triethynylbenzene and 1,4-diethynylbenzene rolled up into nanoparticles with a relatively lower specific surface area of 330 m^2^g^−1^ (Figure 6 FCMP-2). The nanotubes excelled over the nanoparticles, with a lower thermal conductivity of 0.021 Wm^−1^K^−1^ and better dispersity in organic solvents. Their well-defined nanomorphology, good dispersity, and highly aromatic molecular structures exerted excellent flame-resistant efficiency of coating materials (Figure 6) [88]. Furthermore, as various functional substituents could be introduced into the reagents of ethylnylbenes, surface properties of such flame-resistant materials synthesized through Sonogashira–Hagihara cross-coupling reaction can be tailored [89].

### 3.5. Polymers in Flame-Retardant Coatings

In contrast to the aforementioned nanomaterials, polymers can be synthesized after coating their monomers onto the surface of substrate materials. Due to the high dispersity of small molecular monomers, the synthesized polymers can be mixed well with other reagents at desired molar ratios. Meanwhile, many chemically inert reagents, such as ammonium polyphosphate, melamine, and layered double hydroxides, can be mixed with monomers in the coating medium and entrapped after in situ polymerization [90]. Because the monomers and flame-resistant reagents can be mixed thoroughly at any ratio and the polymerization rates are controllable, the polymer network can be topologically expanded. Thus, such a method is suitable for large-scale production and refinement of building materials. Among all the polymerization methods, photoinitiated polymerization of ethylene moieties is the most widely employed. For example, Huang et al. mixed (N, N-bis (2-hydroxyethyl acrylate) aminomethyl phosphonic acid diethyl ester and melamine thoroughly and painted the solution onto the surface of flammable polyurethane foams. Then, using 2-hydroxy-2-methyl-1- phenyl-1-propanone (i.e., Darocur 1173) as a photoinitiator, a coating film of 25 μm was prepared. With the optimal formula, the coating material efficiently increased the LOI value of polyurethane foams from 18.2% to 24.8% and reduced the pHRR from 359.9 kWm^−2^ to 200 kWm^−2^ [91]. Wang et al. employed trimethylolpropane triacrylate, vinyltriethoxysilane, and perfluorododecanethiol to make a polymer network with the photoinitiator 2-hydroxy-2-methylpropiophenone. The polymer network facilely entrapped premixed 9, 10-dihydro-9-oxa-10-phosphaphenanthrene-10-oxide (DOPO), a well-known reagent with excellent flame-resistant efficiency but relatively inert to chemical immobilization [92]. Nonetheless, Chen et al. chemically linked a small number of DOPO groups onto cellulose derivatized by acrylate substituents. The functionalized cellulose was further polymerized with a photoinitiator to generate a flame-retardant film. The film maintained transparency, as it contained a meager amount of DOPO (i.e., 5 w%), resulting in the instant self-extinguishing of the substrate (i.e., less than 1 s) after ignition [93].

Although the ethylene polymerization method is facile and efficient, photoinitiators need to be mixed thoroughly with monomers in the formulas. A modest content of photoinitiators would result in a dense color of the synthesized coating materials. Meanwhile, the remaining photoinitiator would accelerate the aging of coating materials under light. Ma et al. developed a new polymerization strategy through thiol–ene click chemistry to avoid such shortcomings, with no photoinitiators required [94]. Since no photoinitiators are involved, cross-coupling reactions can also be adopted to synthesize colorless and durable polymer coating materials. The reported methods include the aforementioned Sonogashira–Hagihara cross-coupling reaction and the cross-coupling reaction between hexachlorocyclophosphazene and phytic acid with a basic catalyst [52,88]. The sol–gel method is also employed in the preparation of flame-retardant coatings. Kundu et al. mixed phosphorylated chitosan and 3-aminopropyl triethoxysilane. The amino groups on silane bonded with phosphorous groups of cellulose, while silanol groups silanized to form a compact layer. The coating material efficiently reduced the pHRR value of fabrics by 30% [95]. Alternatively, Li et al. adopted ionic bonds to prepare flame-resistant coatings. Specifically, thermoplastic phenolic resin was coupled with aluminum hypophosphite, and deprotonated phenolic groups on resins were bridged with aluminum cations through ionic bonds. Although the material coated on polystyrene elevated its LOI value from 17% to 27.5%, it is worth mentioning that the ionic nature of such coating materials is less compatible with the surface of hydrophobic substrates. Thus, the coated materials need to be curated at 60 °C for two days to stabilize [96].

## 4. Processing Techniques of Flame-Retardant Coatings

A variety of processing techniques have been reported for preparing flame-retardant coatings, such as chemical vapor deposition [97], layer-by-layer assembly [98], hot pressing [63], painting [45], spraying, electrospinning [99], etc. Among all the techniques, the layer-by-layer assembly (LBL) technique is advantageous because compact films of complicated compositions can be generated, and the arrangement and thickness of layered microstructures can be fine-tuned [98]. In most cases, LBL is based on Coulomb forces between cationic layers and anionic layers. Nonetheless, the appropriate pH of solutions is crucial for the ionization of both layers to form the LBL structure. In such cases, ionizable polymers, such as polyethylene imine [75], alginate [100], chitin [101], and graphene oxide [101], are often employed. Specifically, spraying or dipping methods are used for LBL. For the dipping method, substrate materials are alternatively placed in solutions of cationic and anionic polymers, and thorough stirring is required to attain surface adsorption equilibrium. After multiple cycles, LBL structures can be prepared. The dipping method is suitable for coating multiple chemicals on a single layer, as a solution containing several reagents can be used. In addition, the dipping method is preferable if a long reaction time is required to form a layered structure, for example, if a silanization reaction or the growth of nanocrystalline is required for the assembly of a layer [102,103]. Moreover, the dipping method is appropriate if solutions of different layers are incompatible. For example, Chu et al. prepared an LBL of layered double hydroxide in an aqueous solution and polydimethylsiloxane dissolved in chloroform through the dipping method [104]. The dipping method is appropriate to fabricate LBL coatings of cationic and anionic polymers in alternative sequences. Lin et al. adopted the dipping method to prepare LBL structures of melamine and phytic acid on the surface of bamboo substrates (Figure 7a) [103]. The spraying method is more time-efficient than the dipping method, and it is typically mandatory that the solution conditions of each layered material be the same. Davesne et al. chose the spraying method to prepare h-BN and polyethylene imine LBL coatings. As both solutions are aqueous with a pH value of 8, only a 5 min time interval was required between spraying of each layered material (Figure 7b) [61]. Bae et al. added flame-retardant regents of expanded graphite, silane, and starch into one solution. The solution was continuously sprayed onto the surface of expanded polystyrene (EPS) beads, while the EPS beads rotated and moved toward the outlet.

The sprays coated the EPS beads multiple times within a certain time interval, and the reagents generated uniform LBL coatings on the surface of the beads (Figure 8) [105]. It is obvious that when multiple flame-retardant reagents are soluble or dispersive under the same solution condition, the spraying method is highly efficient and suitable for continuous manufacturing. On the contrary, contrastively different solutions are unsuitable to be prepared by the spraying method. Guo et al. chose the spraying method to prepare LBL coatings consisting of alternating layers of alkylammonium functional silsesquioxane (A-POSS)/phytic acid (PA) complex and titanium oxide@polydimethylsiloxane (TiO_2_@PDMS) composite. Therein, A-POSS/PA complex was suspended in an aqueous solution, but TiO_2_@PDMS was prepared in chloroform. Because the two solutions were incompatible, between spraying of each layered material, the substrate needed to be dried completely at 50 °C for 5 h [51].

Besides the LBL technique, hot pressing is another technique available to prepare flame-retardant coatings. Specifically, reagents of coating materials are mixed thoroughly and transferred into a mold. The thickness of the coating is adjusted by the interval space of the mold. The hot-pressing technique is suitable for materials with high specific surface areas. Ideally, the filling medium should be slimy and contain all the involved reagents. After hot pressing, the coating materials need to be aged under appropriate humidity to enhance their durability. For example, Wang et al. made a mixture of montmorillonite (MMT), graphene oxide (GO), and sodium carboxymethylcellulose (CMC) (Figure 9). Using the hot-pressing technique, ternary mimic-nacre coating layers were formed on the surface of a substrate. In such a design, MMT and GO had well-defined nanomorphology and high specific surface areas. Meanwhile, the solution was slimy, and different reagents could adhere to each other through strong hydrogen bonding. As such, hot pressing was a suitable technique [63].

## 5. Concluding Remarks

In conclusion, novel flame-retardant coatings of building materials can be transformed into compact coke beds, placing a barrier on the burning interface to block the self-sustaining mechanism of combustions. Although a large variety of flame-retardant coatings have been introduced into building materials, it is mandatory to rationalize the design of chemical composition and micromorphology of the coating materials to enhance their durability and flame-resistance efficiency. Basically, the designs obey the bottom-up strategy. According to the specific surface properties of different building materials, reagents that can be adsorbed through hydrophobic interactions and hydrogen bonding are often selected. On the other hand, we need to keep in mind that various reagents of coating materials need to be mixed evenly to enhance their flame-retardant efficiency. Strategies include careful selection of organic medium and well-defined morphology of reagents, nanomaterials, and their compatibilities. In recent years, reported flame-retardant coatings have generally shown excellent flame-retardant properties. Such coatings have also exhibited multiple additional functions, such as waterproof properties, super hydrophobocity, self-cleaning properties, electromagnetic interference shielding properties, intelligent alarming functions, etc. Nonetheless, unlike flame-retardant additives in building materials, coating materials are directly exposed to external environments. Therefore, to accomplish practical applications of such functionalized building materials, it is critical to evaluate their weathering properties and long-term durability. Furthermore, although the release of harmful flame-retardant additives into the indoor environment is a known issue, the release of volatiles from degraded coating materials exposed directly to air and light in external environments has not been investigated to date. As the public is increasingly concerned by the environmental hazards of building materials, it is urgent to systematically study the stability and degradation dynamics of flame-retardant materials under various environmental conditions.

## Figures and Tables

**Figure 1 molecules-28-01842-f001:**
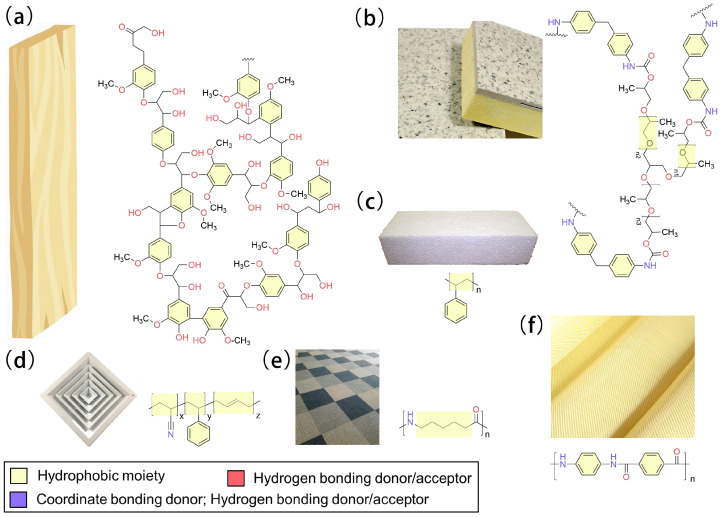
Typical flammable building materials and their molecular structural features that enable the adhesion of flame-retardant coatings: (**a**) wood; (**b**) polyurethane foam; (**c**) polystyrene foam; (**d**) ABS plastics; (**e**) nylon; (**f**) aramid fiber cloth.

**Figure 2 molecules-28-01842-f002:**
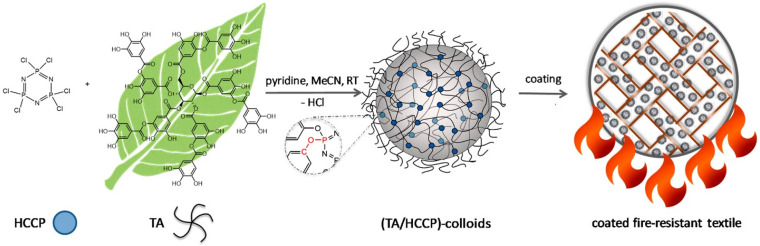
Synthesis route to bio-based (TA/HCCP) colloidal particles through the chemical coupling of hexachlorocyclophosphazene (HCCP) and tannic acid (TA) and the formation of a uniform flame-retardant coating on the textile surface. Reprinted with permission from Ref. [52] Copyright 2020 American Chemical Society.

**Figure 3 molecules-28-01842-f003:**
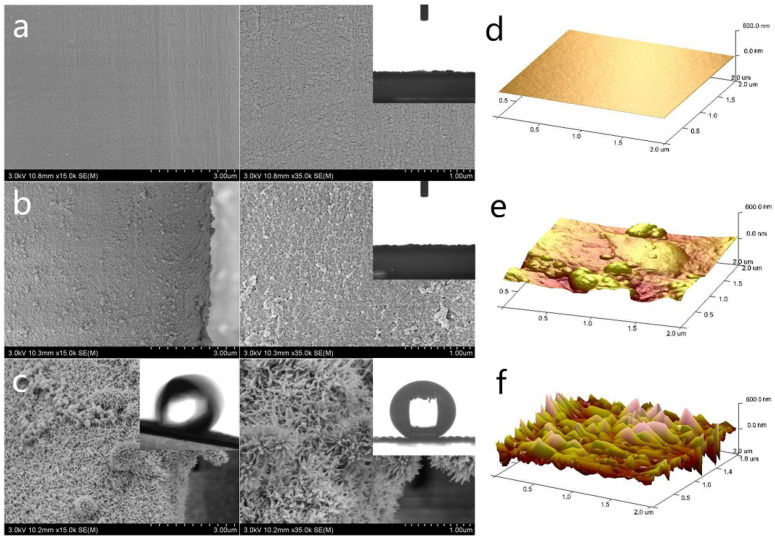
SEM images and AFM images of the silk surface of plain silk fabrics (SF) (**a**,**d**), a silk surface loaded with polydopamine synthesized by laccase catalytic oxidation of dopamine (PDA-SF) (**b**,**e**), and needle-like γ-FeOOH crystal grown in situ on a silk surface loaded with polydopamine (PDA-Fe-SF) (**c**,**f**). The insets in a-c show the corresponding contact angle and sliding angle of the silk fabrics. Reprinted with permission from Ref. [60] Copyright 2020 Elsevier.

**Figure 4 molecules-28-01842-f004:**
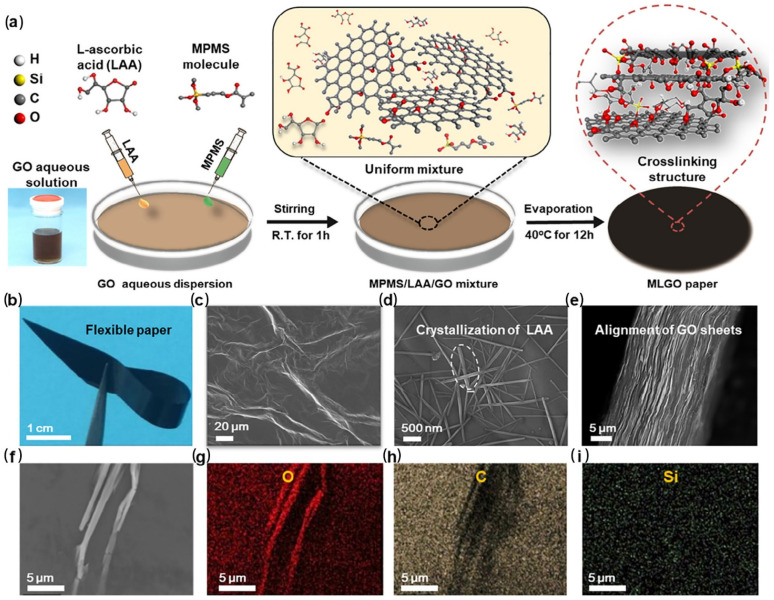
Fabricating process and microstructure of modified multiple-layer graphene oxide (MLGO) paper. (**a**) Schematic preparation of MPMS/LAA comodified GO (MLGO) paper. (**b**) Typical digital photograph of MLGO paper showing good mechanical flexibility. (**c**–**e**) Surface and cross-sectional SEM images of MLGO (10/40) paper. Some needle-like structures are observed on the paper surface, and the GO sheets are well aligned. (**f**–**i**) Typical SEM image and surface element distribution of MLGO (10/40) paper. Reprinted with permission from Ref. [72] Copyright 2020 Elsevier.

**Figure 5 molecules-28-01842-f005:**
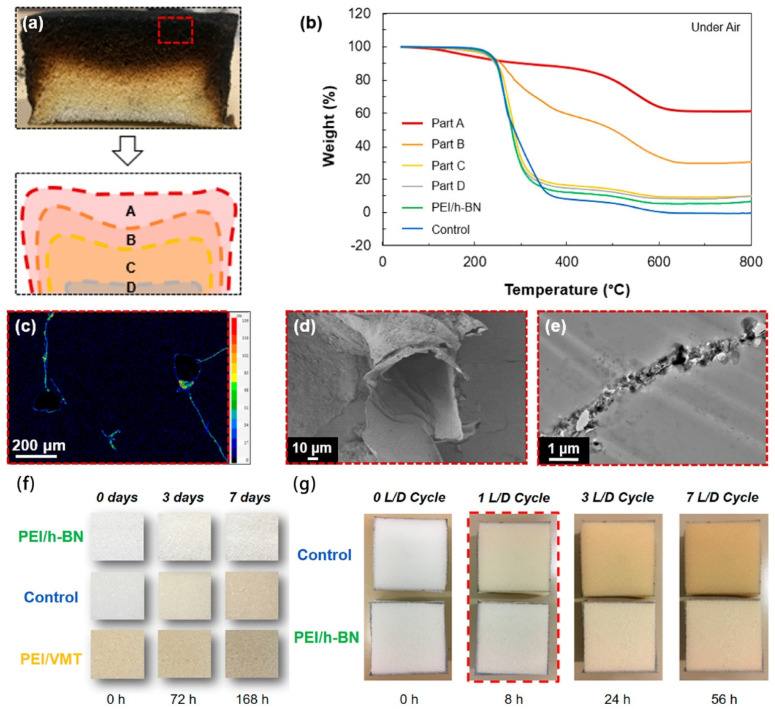
Hexagonal boron nitride nanosheet flame−retardant coating. (**a**) Cross−sectional image of the PEI/h−BN−coated PUF after torch testing with the positions of the four parts (A−D) of the residue studied by (**b**) thermogravimetric analysis compared to uncoated and PEI/h−BN unburnt coated PUF. (**c**) EPMA B X−ray mapping, (**d**) SEM image, and (**e**) TEM cross−sectional micrograph of the charred portion of PEI/h−BN−coated PUF after torch testing. Aging results from (**f**) the natural environment and (**g**) under UV light. Reprinted with permission from Ref. [61] Copyright 2019 American Chemical Society.

**Figure 6 molecules-28-01842-f006:**
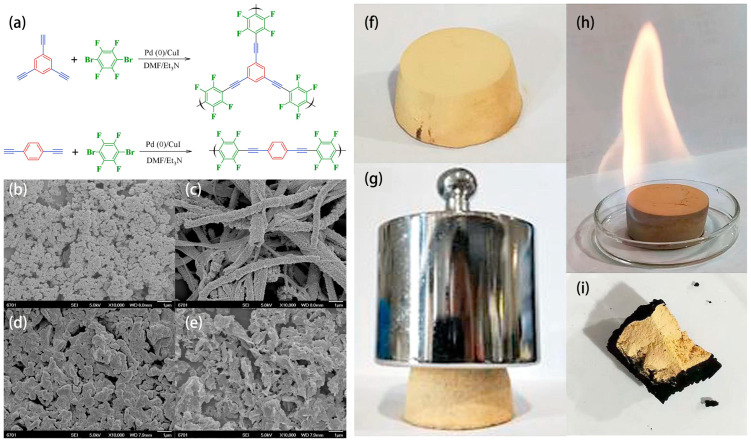
Two flame−retardant coating materials (FCMP-1 and FCMP-2) prepared through Sonogashira–Hagihara coupling (**a**) synthesis route and the morphologies of FCMP-1 and FCMP-2 after soaking in toluene or DMF (**b**) FCMP-1 in toluene, (**c**) FCMP-1 in DMF, (**d**) FCMP-2 in toluene, or (**e**) FCMP-2 in DMF. (**f**) Image of foam coated with FCMP-1 and (**g**) the same foam under 500 g weigh. (**h**,**i**) Image of the foam soaked with ethanol under burning and the residue after burning. Reprinted with permission from Ref. [88]. Copyright 2018 Elsevier.

**Figure 7 molecules-28-01842-f007:**
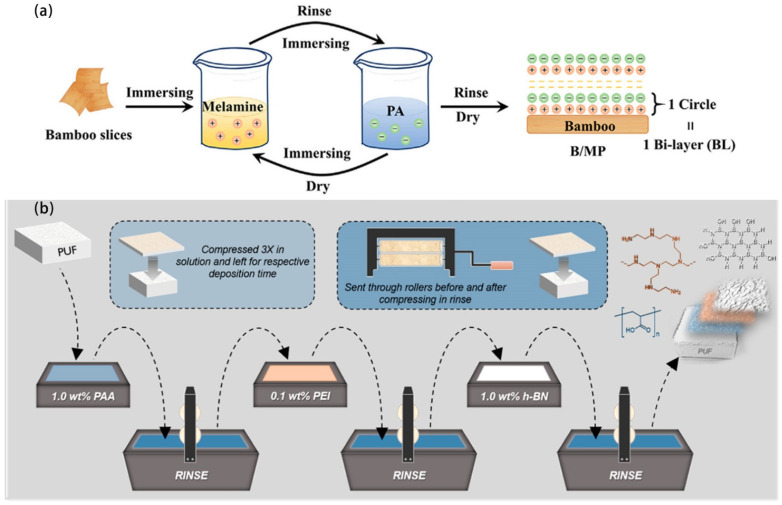
The preparation process of a flame-retardant coating through layer-by-layer assembly by (**a**) dipping method: coating melamine (M) and phytate (P) onto bamboo (B)); reprinted from Ref. [103] and (**b**) spraying method: coating hexagonal boron nitride (h-BN) and polyethylenimine (PEI) onto poly(acrylic acid) (PAA) modified polyurethane foam (PUF)). Reprinted with permission from Ref. [61] Copyright 2019 American Chemical Society.

**Figure 8 molecules-28-01842-f008:**
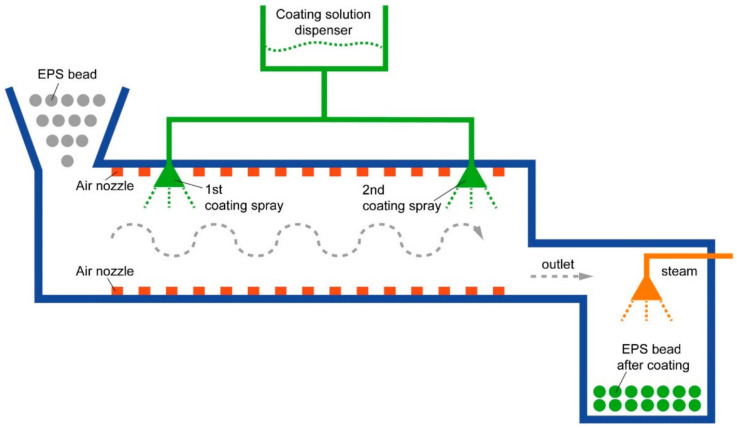
Flow chart of continuous spraying of flame−retardant materials onto the surface of molten expandable polystyrene (EPS) bead microspheres. Reprinted from Ref. [105].

**Figure 9 molecules-28-01842-f009:**
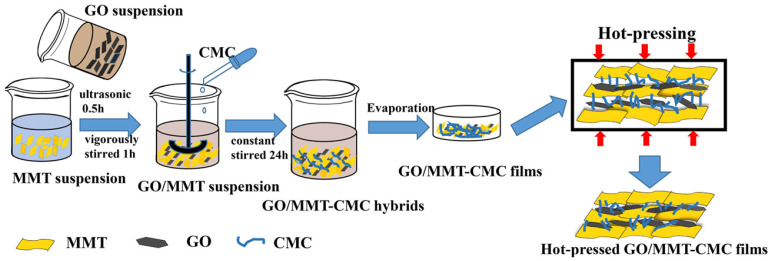
**The** preparation process of ternary mimic−nacre flame−retardant coating based on graphene oxide/montmorillonite−carboxymethycellulose (GO/MMT−CMC) nanocomposite through evaporation assembly and hot pressing. Reprinted with permission from Ref. [63] Copyright 2019 Elsevier.

## Data Availability

Not applicable.

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
