# Peer review of "Comprehensive Review of Recent Research Advances on Flame-Retardant Coatings for Building Materials: Chemical Ingredients, Micromorphology, and Processing Techniques"

_molecules, 2023, doi:10.3390/molecules28041842_

Round 1

Reviewer 1 Report

The paper entitled “Comprehensive Review of Recent Research Advances on Flame Retardant Coatings for Building Materials: Chemical Ingredient, Micromorphology, and Processing Technique” is dealing with a very important part of efficient construction development, that of the fire retardant coatings for this industry. However, in order to be interesting for the Molecules readership, the authors should resolve several issues:oweverH

1.       Please detail the mechanism by which the molecular structural features described in figure 1 can act as anchoring points for different polymers used as coating matrices (e.g. epoxy, polyurea, polydopamine)

2.       The figures should be better explained (e.g. the SEM and TEM images and TGA curves comprised in Figure 5, also Figure 8)

3.       Figure 9 does not appear mentioned in the text.

4.       Overall only the LOI and pHRR values were analyzed, however also the adherence to substrates / mechanical properties and  amount of the fire retardant agent should be emphasized when polymer composite coatings are discussed. 

Author Response

Response to Reviewer:

The paper entitled “Comprehensive Review of Recent Research Advances on Flame Retardant Coatings for Building Materials: Chemical Ingredient, Micromorphology, and Processing Technique” is dealing with a very important part of efficient construction development, that of the fire retardant coatings for this industry. However, in order to be interesting for the Molecules readership, the authors should resolve several issues:

Thank you for your evaluation on the manuscript and constructive suggestions. I have taken your comments very carefully and answer in detail as follows.

Q1: Please detail the mechanism by which the molecular structural features described in figure 1 can act as anchoring points for different polymers used as coating matrices (e.g. epoxy, polyurea, polydopamine)

A1. Thank you for the comments. Coating matrices contain hydrophobic skeleton, hydrogen bonding donor or acceptor, and metal ions can be adhered onto surface of materials. The manuscript has been revised in Page 4: Line 139-143: “Considering the surface properties of all the discussed building materials, it is envisaged that the chemistry of hydrogen bonding, coordinate bonding, and hydrophobic interactions are essential to afford strong affinity between the surface of building materials and flame-retardant coating materials (e.g., epoxy, polyurea, polydopamine, etc.).” I have revised Figure 1 by coloring specific moieties in the molecular structures of typical constructive materials. For example, the hydrophobic moieties are labelled in beige. The hydroxyl groups or ether moieties that could be hydrogen bonding donor or acceptor are marked in light red. The nitrogen atoms contain electron pair could coordinate with metal ions, while they can also be hydrogen bonding acceptors are marked in purple.

Q2: The figures should be better explained (e.g. the SEM and TEM images and TGA curves comprised in Figure 5, also Figure 8)

A2. Thank you for the comment. Following your advices, I have added explanations for Figure 5 in Page 10 Line 379-385: “Davescne et al. made flame-retardant coatings with h-BN nanoplatelets and polyethylene imine, wherein coordination bonds between the boron of h-BN and nitrogen of imine enabled the formation of compact double layers (Figure 5) [61]. Even though the coating material accounted merely ca. ~6.8w% of the substrate (i.e., PU form), it effectively reduced its pHRR value by more than 50%. X-ray mapping, SEM and TEM images further showed that the thin coating material maintained the open-cell structure of the PU foam substrate in the char after combustion (Figure 5 (c-e)).”

I have also added detailed explanations to Figure 8, in Page 14 Line 551-556 “Bae et al. added flame-retardant regents of expanded graphite, silane, and starch into one solution. The solution was continuously sprayed onto the surface of expanded polystyrene (EPS) beads, while the EPS beads rotated in the case and moved toward the outlet. The sprays coated the EPS beads multiple times with a certain time interval, and the reagents generated uniform LBL coatings on the surface of the beads (Figure 8) [105].”

In addition, detailed explanations to SEM images in Figure 4 is also added in Page 9, Line 356-359: “SEM images show that the thin coating sheet exhibited a wrinkled structure, containing evenly distributed needle crystalline of LAA crystalline and tightly aligned graphene oxide sheets (Figure 4 (c-e)).”

Q3: Figure 9 does not appear mentioned in the text.

A3. Thank you for point it out. I have added in Page 15, Line 577:” For example, Wang et al. made a mixture of Montmorillonite (MMT), graphene oxide (GO), and sodium carboxymethycellulose (CMC) (Figure 9).”

Q4: Overall only the LOI and pHRR values were analyzed, however also the adherence to substrates / mechanical properties and amount of the fire retardant agent should be emphasized when polymer composite coatings are discussed.

A4. Thank you for the important comment. I have added more information of the loading amounts of flame-retardant coating materials in the state-of-art designs.

For example, in Page 4 Line 172-175 “For example, Ahmad introduced up to 4w% ground wollastonite into intumescent flame-retardant coating materials, increasing the char expansion by 34%.”

In Page 6 Line 241-243: “The coating material accounting for 12w% of the substrate, significantly elevated the LOI value of cotton to 35%.”

In Page 7, Line 279-281: “The coating layer accounting for 13.7w% of the substrate efficiently increased the LOI value of pure cotton from 17.5% to 26.5%”

In Page 8, Line 319-321:” The three-layer coating grafted on polyurethane foams accounted for merely ~10w% of the substrate, but substantially improved their flame-retardant properties, reducing the pHRR by 54% and restricting the smoke release by 59%.”

In Page 10, Line 381-383: “Even though the coating material accounted merely ca. ~6.8w% of the substrate (i.e., PU form), it effectively reduced its pHRR value by more than 50%.”

The same page, Line 389-391: “The coating material accounting for 6.1~14.4w% of the PU foam substrate could effectively reduce the pHRR value of PU foams by 50.1% and restrict CO release by 53.8%”

The same page, Line 414-417: “By introducing merely 2w% such coating materials onto the substrate thermoplastic PU, the coating materials rapidly generated char and MXene jammed networks, blocking heat and mass transfer and restricting the release of volatile products in combustion [82].”

On the other side, it is somewhat frustrating that most reported work did not discuss on the adherence of flame-retardant coatings towards the substrates, or its mechanical properties. It is found that Davescne et al. have investigated the adherence of coatings on the substrate in details with X-ray mapping, SEM and TEM images. I agree that the adherence between coating materials and substrate is not only pivotal in its flame-retardant efficiency but also important for its safety and long-term stability. As has been discussed in the concluding remarks in Page 16 Line 611-617: “Although the release of harmful flame-retardant additives into the indoor environment is a known issue, it has never been investigated the release of volatiles from degraded coating materials that exposed directly to air and light of external environments. As the public is increasingly concerned by the environmental hazards of building materials, it becomes urgent to systematically study the stability and degradation dynamics of flame-retardant materials under various environmental conditions.”

Reviewer 2 Report

In this work, the authors reviewed the recent development of fire-retardant coatings. This topic is interesting, and the manuscript is well-organized, but the uploaded manuscript still needs some revisions before publication. The specific issues are listed as follows.

1.     In introduction, ‘Thereby, the heat feedback mechanism…quench radical chain reactions’, some references should be cited to support these points, such as Journal of Materials Science & Technology, 2022, 112, 315-328 and European Polymer Journal, 2022, 180: 111581.

2.     In the section 3.1, the properties of the reported coatings should be indicated, not just the components. In addition, some references on this topic should be cited, such as Frontiers in Materials, 2021, 8, 712188 and Colloids and Surfaces A: Physicochemical and Engineering Aspects, 2022, 655, 130292.

3.     Many kinds of fire-retardant coatings were reported in this work, the authors should compare their performances and indicate which coating performs better.

4.     The English of this manuscript should be polished.

5.     Please double check the reference format.

Author Response

Response to Reviewer:

In this work, the authors reviewed the recent development of fire-retardant coatings. This topic is interesting, and the manuscript is well-organized, but the uploaded manuscript still needs some revisions before publication. The specific issues are listed as follows.

Thank you for your interest in the reviewing topic and evaluation on the review paper. I have taken your comments and constructive suggestions very carefully, and answer in detail as follows.

Q1: In introduction, ‘Thereby, the heat feedback mechanism…quench radical chain reactions’, some references should be cited to support these points, such as Journal of Materials Science & Technology, 2022, 112, 315-328 and European Polymer Journal, 2022, 180: 111581.

A1. Thank you for your suggestions. The references are cited in the manuscript as Ref. [6] and Ref. [7] at appropriate positions.

Q2: In the section 3.1, the properties of the reported coatings should be indicated, not just the components. In addition, some references on this topic should be cited, such as Frontiers in Materials, 2021, 8, 712188 and Colloids and Surfaces A: Physicochemical and Engineering Aspects, 2022, 655, 130292.

A2. Thank you for your suggestions. The references are cited in the manuscript as Ref. [35] and Ref. [42] at appropriate positions.

I have also discussed on the properties of the intumescent flame-retardant coatings in the section, as added in Page 3, Line 172-176: “For example, Ahmad introduced up to 4w% ground wollastonite into intumescent flame-retardant coating materials, increasing the char expansion by 34%. Meanwhile, it was found that a longer grinding time and a higher amount of the ground wollastonite ingredient improved the thermal properties of the intumescent coating [40].”

The same page, Line 184-186: “Abdullah et al. found that rice husk ash as an ingredient increased the total and open porosities and rough surfaces of the coating material, improving its intumescent flame-retardant performances [44].”

The same page, Line 188-192:” The nano-composite was prepared as nano-fillers in flame-retardant coatings, increasing the LOI value of butadiene-acrylonitrile rubber substrates from 23% to 28.2%. In addition, the loaded iron could also catalyze the carbonation of polymers during combustion to form a coke bed on the surface and rapidly stop fires. [46].”

Q3: Many kinds of fire-retardant coatings were reported in this work, the authors should compare their performances and indicate which coating performs better.

A3. Thanks for the suggestion. We have actually made comparisons based on the flame-retardant performances and loading amounts of the designed coating materials. Based on the comparison among the reported coating materials, regardless of the costs and other additional functions, it is concluded that “we need to keep in mind that various reagents of coating materials need to be mixed evenly to enhance their flame-retardant efficiency. The strategies include careful selection of organic medium, well-defined morphology of reagents, nanomaterials, and their compatibilities.”

Q4: The English of this manuscript should be polished.

A4. Thanks for the comment. The manuscript has been polished thoroughly. The revisions are highlighted in the manuscript and listed as follows.

Line 29-30 Revision is made: “Furthermore, in recent decades, urbanization in major populous nations has led to mas-sive densification of houses.”

Line 47: Revision is made: “Specifically, such flame retardants quickly decompose and absorb heat released from combustion, emitting non-flammable molecules, such as halogen hydrides.”

Line 54: Revision is made: “Specifically, flame retardants are thermal-stable”

Line 64: Revision is made: “increased usage of flame-retardant additives and their continuous release from building materials into indoor environments are found to be persistent”

Line 79: Revision is made: “Here, the chemical composition, physical micromorphology …”

Line 106: Revision is made: “the downstream products of the petroleum industry”

Line 126: Revision is made: “However, high-performance textile building materials are also highly flammable”

Line 130: Revision is made: “the products could not form compact coke. Thus, they are unable to block mass and heat transfer…”

Line 152: Revision is made: “The most favorable flame-retardant minerals…”

Line 160: Revision is made: “which could easily disperse in flame-retardant coatings”

Line 170: Revision is made: “most natural minerals are bulky crystalline, and unable to suspend in organic mediums”

Line 215: Revision is made: “silsesquioxane (A-POSS) and anionic phytic acid (PA), complexed through ionic forces and adsorbed through hydrogen”

Line 224-226: Revision is made on the caption Fig. 2: “Synthesis route to bio-based (TA/HCCP)-colloidal particles through the chemical coupling of hexachlorocyclophosphazene (HCCP) and tannic Acid (TA) and formation of a uniform flame-retardant coating on the textile surface.”

Line 326: Revision is made: “Instead of the layer-by-layer assembly (LBL) technique…”

Line 360-363 rephrased: “It is worth mentioning that a loose assembly of graphene oxide layers is prepared in such a design. This is contrastively different from conventional designs of flame-retardant coatings aiming to synthesize compact layered nanostructures.”

Line 366: Revision is made: “the coating materials before and after combustion is pivotal in designing fire-alarm intelligent coatings.”

Line 380: Revision is made: “between the boron of h-BN and nitrogen of imine”

Line 412: Revision is made: “hydrogen bonding- induced self-agglomeration”

Line 489: Revision is made: “Wang et al. employed trimethylolpropane triacrylate, vinyltriethoxysilane, and per-fluorododecanethiol to make …”

Line 504: Rephrased “Ma et al. developed a new polymerization strategy through thiol-ene click-chemistry to avoid such shortcomings, …”

Line 506: Rephrased “Since no photoinitiators are involved, cross-coupling reactions could also be adopt-ed to synthesize colorless and durable polymer coating materials.”

Line 524: typo corrected.

Line 560: Revision is made: “LBL coatings consisting of alternating layers…”

Line 606: typo corrected.

Q5: Please double check the reference format.

A5. Thank you for the suggestion. I have double checked the reference format.